# Identification and Expression Analysis of MPK and MKK Gene Families in Pecan (*Carya illinoinensis*)

**DOI:** 10.3390/ijms232315190

**Published:** 2022-12-02

**Authors:** Juan Zhao, Kaikai Zhu, Mengyun Chen, Wenjuan Ma, Junping Liu, Pengpeng Tan, Fangren Peng

**Affiliations:** 1Co-Innovation Center for Sustainable Forestry in Southern China, Nanjing Forestry University, Nanjing 210037, China; 2College of Forestry, Nanjing Forestry University, Nanjing 210037, China

**Keywords:** pecan, MPK, MKK, gene family, expression, organ

## Abstract

Mitogen-activated protein kinases consist of three kinase modules composed of MPKs, MKKs, and MPKKKs. As members of the protein kinase (PK) superfamily, they are involved in various processes, such as developmental programs, cell division, hormonal progression, and signaling responses to biotic and abiotic stresses. In this study, a total of 18 MPKs and 10 MKKs were annotated on the pecan genome, all of which could be classified into four subgroups, respectively. The gene structures and conserved sequences of family members in the same branch were relatively similar. All MPK proteins had a conserved motif TxY, and D(L/I/V)K and VGTxxYMSPER existed in all MKK proteins. Duplication events contributed largely to the expansion of the pecan MPK and MKK gene families. Phylogenetic analysis of protein sequences from six plants indicated that species evolution occurred in pecan. Organ-specific expression profiles of MPK and MKK showed functional diversity. *Ka*/*Ks* values indicated that all genes with duplicated events underwent strong negative selection. Seven *CiPawMPK* and four *CiPawMKK* genes with high expression levels were screened by transcriptomic data from different organs, and these candidates were validated by qRT-PCR analysis of hormone-treated and stressed samples.

## 1. Introduction

The pecan (*Carya illinoinensis* [Wangenh.] K. Koch) is an important economic tree species of the genus *Carya* in the family Juglandaceae, native to the United States and northern Mexico, and is one of the world’s most famous dried fruit tree species [1,2]. The oil content of the pecan nut is about 70%, which is an excellent woody oil tree species. The nuts contain a lot of healthy unsaturated fatty acids and high levels of antioxidants, as well as some phytochemicals chemical substances, such as phenolic compounds [3,4,5]. Studies have shown that eating these nuts can reduce the incidence of various diseases, such as tumors, edema, and hyperlipidemia, while providing the dietary fiber, protein minerals, and vitamin B needed by the human body [6,7].

Plants are frequently challenged by a variety of biotic and abiotic stresses, including pathogen infection, drought, salt, cold, and oxidative stress. These unfavorable conditions adversely affect plant growth and development, and lead to a substantial reduction in crop yield. Plants have evolved into a series of extremely complex signaling networks and regulatory mechanisms to defend themselves against these adverse conditions [8,9,10].

Mitogen-activated protein kinases consist of three kinase modules composed of MAPKs (MPKs), MAPK Kinases (MKKs), and MAPKK Kinases (MAPKKKs/MEKKs) [11,12,13]. MKKs are activated by MPKs when serine and threonine residues in the S/T-xxxxx-S/T (S: serine; x: any amino acid; T: threonine) motif are phosphorylated. In turn, MKKs can activate MPKs when conserved threonine and tyrosine residues in the TxY (T: threonine; x: any amino acid; Y: tyrosine) motif are phosphorylated [14,15,16,17,18]. Protein kinases are important mediators of internal and external signal exchange in cells [19]. Studies have shown that the *MAPKKK5-MKK4/5-MPK3/6* pathway is very important for chitin signaling in Arabidopsis [20]. Activated MAPKs phosphorylate downstream substrates, affecting the biochemical properties of cells and leading to appropriate export responses [21,22], and they can be divided into four major categories based on the presence of a TEY or TDY motif in the phosphorylation site [23]. As members of the protein kinase (PK) superfamily, they catalyze the phosphorylation of target proteins, resulting in conformational changes that alter their activity, binding, stability, and cellular localization [24,25]. MPKs have many cellular targets that regulate metabolism and transcription [26,27,28]. In order to regulate plant development and respond to environmental stress, plants have evolved to have complex mechanisms to sense and transmit environmental information. MPK cascades are involved in many processes, such as developmental programs [29], cell division [30], hormonal progression [31], and signaling responses to various forms of biotic and abiotic stresses [32,33,34,35].

Many members of the MPK and MKK cascades have been identified using functional genomics approaches. A total of 20 MPKs and 10 MKKs have been found in *Arabidopsis thaliana* [36]. Fifteen MPKs and 8 MKKs have been found in the *Oryza sativa* genome [37]. In addition, a large number of MPK and MKK cascades from poplar (*Populus trichocarpa*) [38], *Brassica napus* [39], maize (*Zea mays*) [40,41], *Brachypodium distachyon* [12], tobacco (*Nicotiana tabacum*) [42], *Ophiocordyceps sinensis* [43], and apple (*Malus domestica*) [44] were successively identified. Although MPK and MKK have been studied in many plant species, no comprehensive analysis of pecan has been performed to date. In the current work, eighteen *CiPawMPK* and ten *CiPawMKK* genes in the pecan were identified, and the evolutionary relationships and duplication events were also analyzed. The organ-specific expression profiles of MPK and MKK were explored The expression patterns of the two gene family members response to different treatments were investigated. The precise annotation of MPK and MKK is the first step to fully understand the roles in pecan phytohormones and stress response.

## 2. Results

### 2.1. Identification, Nomenclature, and Protein Sequence Analysis of MPK and MKK Gene Families

Based on bioinformatics methods, 18 MPK and 10 MKK genes were identified from the whole genome of pecan, and each gene was named according to its homology with *Arabidopsis thaliana*. Genome sequence analysis results showed that the MPK and MKK proteins of the pecan were very conserved. As shown in Appendix A, the longest protein, CiPawMPK9-1, encoded 630 amino acids with a molecular weight of 71.26 kD; the shortest protein, CiPawMKK9-1, encoded only 315 amino acids with a molecular weight of 35.4 kD. The isoelectric point (pI) ranged from 5.42 to 9.31. Subcellular localization prediction results showed that most MPK and MKK proteins were located in the cytoplasm, CiPawMPK9-1 and CiPawMKK6-1 were located in the outer membrane, and three proteins (CiPawMPK16-1, CiPawMPK20, and CiPawMKK2-2) were located in the periplasm.

### 2.2. Phylogenetic Relationship of MPK and MKK Cascade Genes

According to the phylogenetic tree of *Arabidopsis*, rice (*Oryza sativa*), pecan, Chinese hickory (*Carya cathayensis*), walnut (*Juglans regia*), and poplar (*Populus trichocarpa*), all the MPK and MKK genes were divided into four groups (A, B, C, and D) (Figure 1). The gene name and ID of each species is listed in Appendix A. Four *CiPawMPK* gene family members were distributed in both groups A and B, and three members (*CiPawMPK1*, *CiPawMPK7-1*, and *CiPawMPK7-2*) were distributed in group C. Group D contained the most MPK cascade kinases, with 10 *OsMPKs* and *PtMPKs*, 10 *CiPawMPKs*, *CcaMPKs*, and *JrMPKs*, and 8 *AtMPKs*. Compared with the number of MPK gene family members, the number of MKK identified in each species was relatively small. Groups A and D contained approximately the same number of MKKs. Coincidentally, only one MKK gene in group C of the six species is compared in this paper. The specific gene distribution of each species is shown in Appendix A.

### 2.3. Conserved Domains, Motifs, and Gene Structure Analysis in MPK and MKK Cascades

To investigate the domain characteristics of the MPK and MKK gene families, multiple sequence alignments of predicted amino acid residues were performed using DNAMAN 9.0 software. The results showed that all MAPK proteins had a conserved motif TxY, and the genes in groups A and B contained CD domain (LH)DxxDE(P)xC (Appendix A). The conserved motifs D(L/I/V)K and VGTxxYMSPER (V: valine; G: glycine; Y: tyrosine; M: methionine; S: serine; P: proline; E: glutamicacid; R: arginine) existed in all MKK proteins, and only the genes in group A contained the S/T-xxxxx-S/T motif (Figure 2).

The results of the MEME website analysis were visualized using Tbtools version 1.098722, and the results showed that the most conserved motif in MPK was motif 1. Unlike the three groups A, B, and C, the MPK gene in group D contained a very large number of conserved motifs—more than 10 (Figure 3aII). The ubiquitous conserved motifs in MKK were motifs 1/2/3/5. Except for *CiPawMKK3*, other MKKs contained similar numbers of conserved motifs. Twenty-five motifs were found in *CiPawMKK3* within the identified range (Figure 3bII).

Gene structure analysis showed that the MPK contained more exons. The MPK in groups A and B both contained 6 exons. The MPK in group C contained the least number of exons and introns. The exons of MPK in group D were all greater than 9 (Figure 3aIII). The MKK in groups A and B had more than 8 exons. The four MKK genes *CiPawMKK5-1/9-1/9-2/10* had only one exon (Figure 3bIII).

### 2.4. Analysis of Cis-Acting Elements in the Promoter Regions of MPK and MKK Genes

To explore the gene regulatory mechanisms of MPK and MKK, we predicted and analyzed cis-acting elements in the promoter sequences of MPK and MKK genes. Three types of cis-acting elements were found, including hormone-responsive elements, elements related to resistance to harsh environments and stress, and elements related to plant growth and development.

Six different types of environmental and stress-related action elements were retrieved, which mainly included the most number of light-responsive elements (Box 4, G-box, TC-rich repeats, GT1-motif, I-box, GATA-motif, and MRE), low-temperature responsive elements (LTR and GA-motif), drought-inducibility elements (MBS), defense and stress-responsive elements (TC-rich repeats), circadian control elements and the regulatory elements related to anaerobic induction (ARE). The hormone-responsive elements mainly included MeJA-responsive elements (CGGTA-motif and TGACG-motif), auxin-responsive elements (AuxRR-core, TGA-element, and TGA-box), gibberellin-responsive elements (P-box, GARE-motif, and TATC-box), salicylic acid-responsive elements (TCA-element), and abscisic acid-responsive elements (ABRE).

The MPK contained 117 hormone-responsive elements, and MKK contained 65, among which the MeJA-responsive elements and salicylic acid-responsive elements had the largest number, indicating that MPK and MKK may play vital roles in the process of fruit ripening and the leaf abscission of pecan. Five cis-acting elements involved in the regulation mechanism of pecan growth and development were retrieved, which mainly included expression elements (CAT-box), seed-specific regulation elements (RY-element), endosperm expression elements (GCN4-motif), wound-responsive elements (WUN-motif) and cell cycle regulation elements (MSA-like), respectively. Only one wound-responsive element was found in the promoter region of MPK, and cell cycle regulation elements were specific to MKK (Figure 4).

### 2.5. Genome Distribution and Gene Duplication of CiPawMPKs and CiPawMKKs

As shown in Figure 5, the MPK and MKK cascade kinase genes were distributed on 13 chromosomes of the pecan. Chr 02 and Chr 03 each had four genes, which were the two chromosomes with the largest number of genes. Three MPKs (*CiPawMPK4-3*, *CiPawMPK6-2*, and *CiPawMPK16-1*) and one MKK (*CiPawMKK5-1*) were found on Chr 02. Moreover, three *CiPawMKKs* and one *CiPawMPK* were distributed on Chr 03. Other MPKs and MKKs were scattered across the remaining eleven chromosomes. These results suggested that the pecan had undergone genetic variation during evolution.

Moreover, the gene collinearity in pecan, between pecan and Arabidopsis, and between pecan and rice were analyzed using MCScanX software. Five duplication gene pairs were found in pecans, twenty-four between pecan and *Arabidopsis*, and five between pecan and rice. All linear relationships are shown in the genomic map (Figure 6).

The synonymous (*Ks*) and non-synonymous (*Ka*) duplication events were further calculated, and the effect of selection pressure on sequence differences was explored using the ratio of *Ka/Ks* (Table 1). *Ka*/*Ks* > 1 means positive selection, *Ka/Ks* = 1 means neutral selection, and *Ka*/*Ks* < 1 means negative selection. Amino acid substitutions that improve fitness were obtained by positive selection, whereas amino acid substitutions that reduce fitness were negative selections [37]. Fragments and linkage events from the MPK and MKK gene families in pecan, *Arabidopsis*, and rice have *Ka/Ks* values ranging from 0.01 to 0.30, suggesting that these genes may experience strong negative selection.

### 2.6. Expression Patterns and Co-Expression Networks of MPK and MKK in Different Organs of Pecan

From the heatmap (Figure 7), we found that except for the low expression level of seeds, the other five organs materials had relatively high expression levels. The data to the log_2_ transformed (FPKM+1) values were concentrated between three and six. Three *CiPawMPKs* (*CiPawMPK4-1*, *CiPawMPK4-2* and *CiPawMPK7-2*) and three *CiPawMKKs* (*CiPawMKK6-1*, *CiPawMKK6-2* and *CiPawMKK10*) showed very low expression levels in six organs. It can be preliminarily concluded that both MPK and MKK were involved in the formation and development of the vegetative and reproductive organs of pecan.

The co-expression network was further constructed, and the interrelationship between pecan MPK and MKK genes was analyzed by the transcription levels of pecan in different organs. In the network, 27 nodes (17 MPK and 10 MKK) and 43 edges were found, and 14 nodes contained more than 6 edges, indicating a strong correlation between these nodes (Appendix A, Appendix A).

### 2.7. Real-Time Quantitative Analysis of CiawMPK and CiPawMKK under Various Phytohormone Treatments and Abiotic or Biotic Stresses

MPK and MKK cascade kinase genes are not only involved in plant growth and development, but also play key roles in controlling plant responses to abiotic, biotic stresses, and phytohormones. In this study, we investigated the changes of MPK and MKK gene expression in pecan under four phytohormone treatments including SA, ABA, MeJA, and 6-BA, one stress treatment H_2_O_2_, and two abiotic stress including NaCl and PEG using qRT-PCR. Our findings were consistent with Chen’s wherein most MPK and MKK genes were induced or constitutively expressed under stress treatment [10]. After 3 h of phytohormone treatment, 55% of the genes were up-regulated, 25% of the genes had the highest expression level at 12 h, and the expression level showed a down-regulation pattern at the time point after reaching the threshold (Figure 8 and Figure 9).

*CiPawMPK3-1* and *CiPawMPK4-3* were up-regulated at 3 h, 6 h, and 12 h after H_2_O_2_ treatment, respectively, and their expression levels began to decrease at 24 h. *CiPawMPK7-1* showed a downward trend at each time point sampled (Figure 10). Seventy three percent of MPK and MKK genes were up-regulated after NaCl and PEG treatment for 8 days. *CiPawMKK2-2* and *CiPawMKK3* were up-regulated after treatment with NaCl. The expression of *CiPawMPK3-1*, *CiPawMPK1*, *CiPawMKK2-1,* and *CiPawMKK5-1* was up-regulated after treatment with PEG, and the expression levels of the remaining MPK and MKK genes began to decrease after 8 d (Figure 11).

### 2.8. Changes of SOD and POD Activity in Pecan Treated with NaCl and 20% PEG

Stress reduces the ability of plant antioxidant systems to scavenge ROS (reactive oxygen species), and long-term accumulation of large amounts of ROS will cause damage to plant growth and development [45]. It is generally accepted that abiotic stress significantly increases the production of ROS [46], antioxidant enzymes are activated, of which SOD and POD play important roles in removing ROS generated under stress [47,48,49,50]. In this study, the SOD activity of pecan was significantly increased after treatment with NaCl and 20% PEG. The POD activity of NaCl-treated seedlings decreased briefly at 8 d, and began to show an upward trend at 16 d. The POD activity of the seedlings treated with 20% PEG reached the maximum value at 8 d, and then began to decline (Appendix A).

## 3. Discussion

Further analysis of gene families requires extensive manual inspection of raw sequences, especially those obtained from non-model genomes. Studies have shown that different MKKs can also act on the same MPK after different external environmental stimuli. For example, in *Arabidopsis*, *AtMPK3* and *AtMPK6* can be regulated by *AtMKK9* in the ethylene signaling pathway [51], but can be phosphorylated by *AtMKK4* and *AtMKK5* after flagellin activation [52].

So far, three *AtMPKs* (*AtMPK3*, *AtMPK6* and *AtMPK10*) in group A have been studied in detail, and they were mainly involved in stress response and hormone signal transduction. Studies have shown that *AtMPK3* and *AtMPK6* were involved in almost all life activities of plants, not only in abiotic stress [53,54,55,56], but were also involved in hormone and growth and development signal transduction pathways [57,58], In particular, its role in plant disease resistance is becoming more and more important [59]. *OsMPK5* is homologous to *NtWIPK* and *AtMPK3*, and will be induced by different pathogens and environmental stimuli [38,60]. *AtMPK4* in group B was inactivated by transposon insertion, so that the mutant material exhibited constitutive systemic acquired resistance [61]. Group C contains four members: *AtMPK1*, *AtMPK2*, *AtMPK7,* and *AtMPK14*. A recent study showed that *AtMPK1* and *AtMPK2* can be activated by mechanical injury, ABA, JA, and also H_2_O_2_-induced activation [62]. The expression of *CiPawMPK7-1* gradually decreased after being induced by 6-BA, ABA and H_2_O_2_, *CiPawMPK7-1*, *AtMPK1* and *AtMPK2* belonged to the same phylogenetic tree group, and they may play similar roles in the regulation process of plant hormones. *AtMKK3* is the upstream interacting component of *AtMPKs* in group C, and it was confirmed that *AtMKK3-AtMPK7* is involved in the regulation of H_2_O_2_, and enhances the activity of *AtMPK7* induced by H_2_O_2_ [63]. *CiPawMKK3* also enhanced the ability of the MPK genes in group C, which was induced by H_2_O_2_.

Few MKK genes have been identified in plants, with only 10 MKKs in *Arabidopsis* and 8 MKKs in rice, which are only half of the MPKs, suggesting that different MPKs may be activated by the same MKK, which is the intersection of the MPK cascade pathway network. It can be seen that the MKK gene plays an important role in the MPK signal transduction pathway. Group A includes *AtMKK1*, *AtMKK2,* and *AtMKK6*. Studies have shown that *AtMKK1* and *AtMKK2* are jointly involved in the innate immune response of plants [64], whereas *AtMKK2* is involved in salt stress and low temperature stress in plants [65], and *NtMKK1* and *AtMKK6* have similar functions, participate in the process of tobacco cell division, and can activate the MPK gene [66].

The conserved CD domain (LH) Dxx(EP)xC is the docking site with MKK in MPK, and the two adjacent D and E residues of this domain play a key role in interacting with alkaline residues K (Lysine) and R (Arginine) in MKK [67]. We found a CD domain in all CiPawMPK of group A and B, but not in groups C and D (Appendix A).

Given that MPK cascade genes are involved in a wide range of physiological processes, their abundance varies widely [68]. Within the five duplicated gene pairs identified in pecan, only two pairs (*CiPawMPK16-1/CiPawMPK* and *CiPawMKK2-1/CiPawMKK2-2*) shared similar expression patterns in nearly all tested organs. Our observations are consistent with previous findings on preferential organ expression in MAPK cascade gene pairs [69,70]. Duplicate gene pairs may differ in expression ability and function among different organs. For example, *CiPawMPK4-3* was highly expressed in other organs except for seeds, whereas the expression of its paralogs *CiPawMPK4-2* is extremely low in the 6 organs sampled in this study (Figure 7).

## 4. Materials and Methods

### 4.1. Genome-Wide Identification of MPK and MKK Gene Family in Pecan

To identify pecan MPK and MKK cascade kinase gene families, all protein sequences were downloaded from the newly published pecan genome (https://phytozome-next.jgi.doe.gov/info/CillinoinensisPawnee_v1_1 (accessed on 10 March 2022)) (Taxonomy ID: 32201) [71]. Twenty AtMPKs and 10 AtMKKs protein sequences were retrieved from the *Arabidopsis* Information Resource (TAIR) database (http://www.arabidopsis.org/ (accessed on 12 March 2022)) [72]. Hidden Markov models (HMM) were built using the hmmbuild program in HMMER version 3.3.2 (http://www.hmmer.org/ (accessed on 14 March 2022)) to search for putative MPK and MKK proteins in pecans with an E-value < 1 × 10^−10^ [73]. The protein sequences were blasted using the NCBI online website (https://blast.ncbi.nlm.nih.gov/Blast.cgi (accessed on 14 March 2022)), based on the phylogenetic pecan tree and *Arabidopsis thaliana*, and redundant sequences were manually deleted, and the final MPK and MKK cascade kinase genes of pecan, Chinese hickory and walnuts, three important economic tree species belonging to the same family of Juglandaceae, were determined. To explore the evolutionary relationship between them, the protein sequences of walnut were downloaded from the walnut genome database (http://xhhuanglab.cn/data/juglans.html (accessed on 15 March 2022)), and the protein sequences of Chinese hickory were retrieved from the GigaScience database (http://gigadb.org/dataset/100571 (accessed on 15 March 2022)) [74]. The MPK and MKK genes contained in both were identified by the same method.

### 4.2. Sequence Characterization Analysis

The molecular weights (MWs), isoelectric points (pIs), and the number of amino acids of MPK and MKK kinase proteins were calculated using the ExPASy online website (https://web.expasy.org/protparam/ (accessed on 18 March 2022)). Subcellular localization information of MPK and MKK proteins was predicted using the CELLO version 2.5 (http://cello.life.nctu.edu.tw/ (accessed on 22 March 2022)) [75].

### 4.3. Multiple Sequence Alignment Analysis

The multiple sequence alignment of pecan MPK and MKK protein sequences were performed using the Clustal X version 2.0 [76], and the images of the conserved domains were visualized with DNAMAN 9.0 software.

### 4.4. Exon-Intron Structure and Conserved Motif Analysis

The conserved motifs in MPK and MKK protein sequences were analysed using the MEME online website (http://meme-suite.org/ (accessed on 2 April 2022)) [77], and the parameters were set to the number of repetitions: any; maximum numbers of motifs: 35; optimum motif widths: 6-200 [78], and the images of the gene structure and motif were visualized using the Tbtools version 1.098722 (https://github.com/CJ-Chen/TBtools (accessed on 4 April 2022)) [79].

### 4.5. Construction and Analysis of Phylogenetic Tree

The protein sequences of MPK and MKK genes of rice and poplar were downloaded from Phytozome database version 13 (https://phytozome-next.jgi.doe.gov/ (accessed on 10 April 2022)) [80] according to the gene IDs reported in the literature [38]. The Clustal Muscle module in MEGA 7 software (https://www.megasoftware.net (accessed on 12 April 2022)) [81] was used to perform the multiple sequence alignment of MPKs and MKKs of pecan, Chinese hickory, walnut, *Arabidopsis*, rice, and poplar, and the neighbor-joining (NJ) method was used to generate the no-root evolutionary tree. Parameter bootstrap settings were checked and repeated 1000 times. Finally, the phylogenetic trees were constructed using the Evolview online website (https://evolgenius.info//evolview-v2/#login (accessed on 16 April 2022)).

### 4.6. Cis-Acting Element Analysis

In order to explore the possible cis-acting elements in the promoters of pecan MPKs and MKKs, the genome sequences of pecan 2000 bp upstream of MPK and MKK genes were extracted using the Tbtools version 1.098722. Then, cis-acting elements were predicted using the Plant CARE online website (http://bioinformatics.psb.ugent.be/webtools/plantcare/html/ (accessed on 4 May 2022)), and the images were visualized using the Tbtools version 1.098722.

### 4.7. Gene Distribution Prediction

The pecan genome has a total of 16 chromosomes (2n = 32). In order to explore the distribution of MPK and MKK on chromosomes, we extracted the chromosome sequence length file of pecan and the chromosomal location files of MPK and MKK, and the interacting gene pairs, were visualized with the Tbtools version 1.098722.

### 4.8. Gene Duplication Analysis and Ka/Ks Value Calculation

The gene collinearity in pecan, between pecan and Arabidopsis, and between pecan and rice were analyzed using the MCScanX software (http://chibba.pgml.uga.edu/mcscan2/ (accessed on 10 May 2022)) [82], with the E-value set to 1 × 10^−5^. First, the coding sequences (CDS) of MPK and MKK genes were aligned according to the protein sequence alignment using the muscle program with default options in MEGA 7. Then, the synonymous substitution (*Ks*) and nonsynonymous substitution (*Ka*) were calculated separately based on multiple sequence alignment using the MEGA version 7, and finally, the *Ka/Ks* of collinear gene pairs were calculated to evaluate selection pressure.

### 4.9. Plant Material, Growth Conditions, Treatments, and Sample Collection

Samples of leaves, mature male and female flowers, young fruits, and seeds were randomly collected from nine-year-old healthy pecan trees of the ‘Pawnee’ variety at the Pecan Experimental Base of Nanjing Forestry University, located in Jurong City, Jiangsu Province, China (119°9′6″ E, 31°52′45″ N). Male flowers are picked in late April, female flowers and leaves are picked in early May, young fruits are the whole fruit picked in July, and the root samples were collected from three-month-old pecan seedlings propagated by seeds and harvested from ‘Pawnee’ trees, with the seeds being mature seeds. All samples were collected on the morning of a sunny day.

Seventy-two annual pecan seedlings with a uniform size and robust growth (average plant height 85 cm, average ground diameter 15 mm) were selected and cultured in a 4:1:1 mixture of peat soil, perlite, and vermiculite. The test site is located in the climate chamber of the Biotechnology Building of Nanjing Forestry University, with a temperature of (25 ± 2) °C and a humidity of 70–80%. For hormone analysis, seedlings were foliar sprayed with 1 mM SA, 100 μM ABA, 100 μM MeJA, and 20 μM 6-BA, leaf samples were collected at 0, 3, 6, 12, and 24 h, and the same experimental design was performed with 10 mM H_2_O_2_. For abiotic stress, seedlings were soiled with 20% PEG and 200 mM NaCl, and leaf samples were collected at 0, 8, 16, and 24 days, respectively.

All collected organ samples were frozen in liquid nitrogen and stored in a laboratory at −80 °C freezer until RNA was isolated. Each sample was collected from three plants and three biological replicates were set up for each treatment.

### 4.10. RNA Extraction and qRT-PCR Analysis

Take 50~100 mg of collected organ samples and put them into centrifuge tubes and then put them in liquid nitrogen for quick freezing, and then use a grinder to grind the samples into powder. The total RNA of different samples was extracted using the FastPure^®^ Plant Total RNA Isolation Kit (Vazyme, Nanjing, China) [83]. Then, RNA quality was measured using an Agilent 2100 bioanalyzer (Agilent Technologies, CA, USA). First-strand cDNA was synthesized using the HiScript^®^Ⅲ RT SuperMix for qPCR (+gDNA wiper) (Vazyme, Nanjing, China) [84], which was stored in a refrigerator at −20 °C.

Specific primers for the *MPK* and *MKK* genes were designed using the IDT PrimerQuest online tool (https://sg.idtdna.com/Primerquest/Home/Index (accessed on 6 November 2022)), and the primers were synthesized at Qingke Biotechnology Co., Ltd. (Nanjing, China) (Appendix A). For qRT–PCR (quantitative real-time PCR) analysis, the actin gene (*CiPaw.03G124400*) was used as an internal control for normalization [85]. The experiments were performed on an ABI 7500 Real Time PCR system (Applied BiosystemsTM, Foster City, USA) using the Taq pro Universal SYBR qPCR Master Mix (Vazyme, Nanjing, China). PCR reaction conditions were as follows: initial denaturation at 95 °C for 30 s, followed by 40 cycles of 5 s at 95 °C and 15 s at 60 °C. Relative quantification of MPK and MKK genes was performed by the 2^−∆∆Ct^ method [86].

### 4.11. Transcriptome and Co-expression Analysis of Pecan in Different Organs

All materials were sent to Gene Denovo Biotechnology Co., Ltd. (Guangzhou, China) to determine the transcriptome information of the samples. Transcriptome data was obtained from the NCBI (National Center for Biotechnology Information) database (https://www.ncbi.nlm.nih.gov/ (accessed on 20 May 2022)) (BioProject ID: PRJNA799663). To assist protein-coding gene annotation, transcriptome data was downloaded from the NCBI database, and the expression level of each protein-coding gene—FPKM (Fragments Per Kilobase of exon model per Million mapped) was calculated using the default parameters of Cufflinks 0.16 software [87]. FPKM > 0.5 was used as the screening standard for gene expression and the FPKM value was used to calculate the transcriptional abundance of the MPK and MKK genes in pecan. The data to the log_2_ transformed (FPKM + 1) values were converted with Microsoft Excel 2016 [88], and the heat maps of MPK and MKK gene expression in different organs of pecan were visualized using the Tbtools version 1.098722.

According to the expression data of MPK and MKK in different organs of pecan, the Pearson correlation coefficient (PCC) value between different gene pairs was calculated by OmicShare (https://www.omicshare.com/ (accessed on 24 May 2022)). A co-expression network was constructed by selecting gene pairs with a PCC greater than 0.7 to explore the relationship between MPK and MKK (Appendix A), and then the network was visualized with Cytoscape version 3.8.0 [89].

### 4.12. Detection of Enzyme Activity

The enzyme solution was extracted with a kit by Suzhou Keming Biotechnology Co., Ltd. (Suzhou, China), and the corresponding OD value of the method was determined, and the activities of SOD (superoxide dismutase) and POD (peroxidase) activity were calculated according to the fresh weight of the sample [90].

## 5. Conclusions

In summary, a total of 18 MPKs and 10 MKKs were annotated on the pecan genome, all of which could be classified into four subgroups. Gene structure and conserved structural domains, chromosomal position, evolutionary relationships, cis-acting elements, duplication events, co-expression networks, and expression patterns of MPK and MKK genes were investigated. The results suggested that *CiPawMPKs* and *CiPawMKKs* may play a role in pecan development and stress as well as in hormonal regulatory mechanisms. Overall, this study provided a basis for further exploration of the functions of pecan MPKs and MKKs.

## Figures and Tables

**Figure 1 ijms-23-15190-f001:**
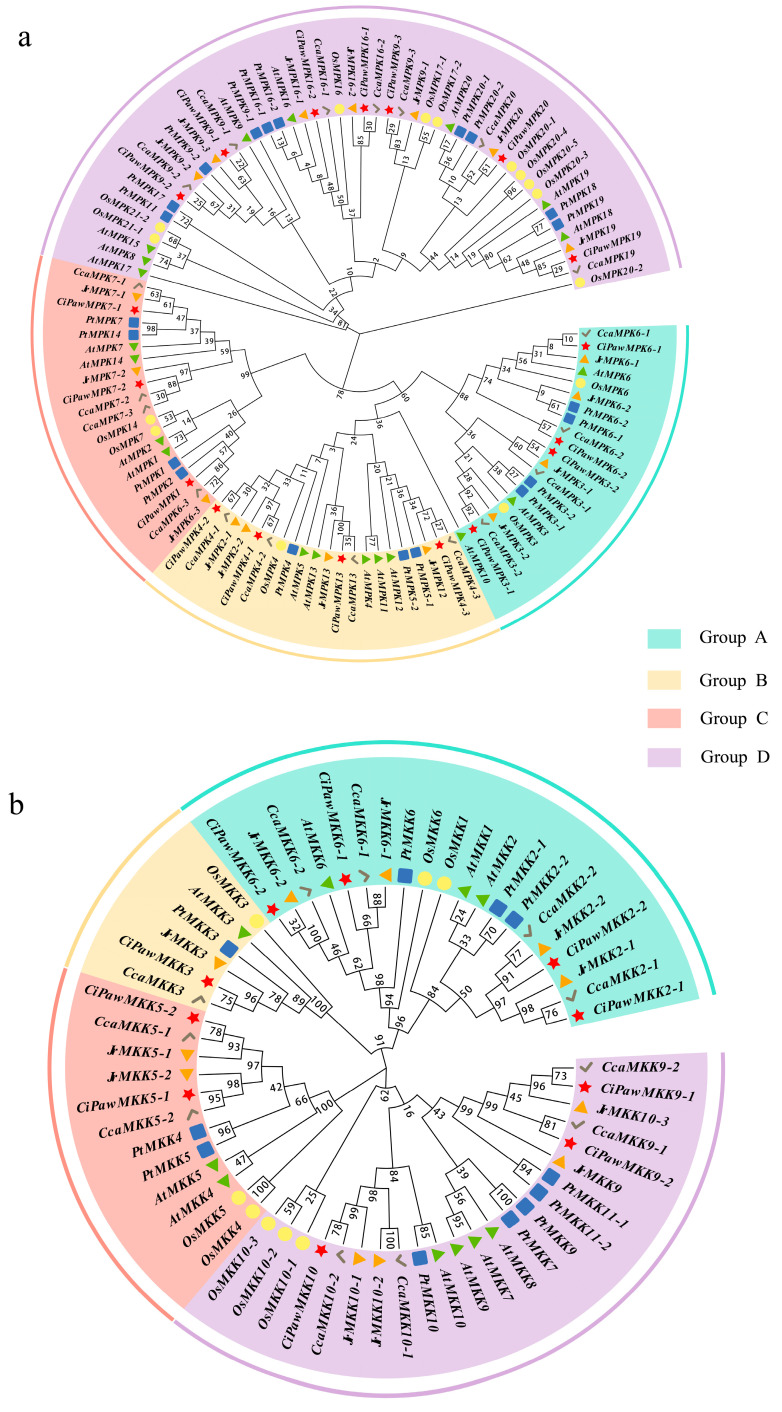
Phylogenetic trees of MPK (**a**) and MKK (**b**) in *Arabidopsis*, rice, pecan, Chinese hickory, poplar, and walnut. The species were distinguished by different colored shapes. Both MPK and MKK from six species were divided into four groups (group A, B, C, and D). The groups were highlighted in different colors.

**Figure 2 ijms-23-15190-f002:**
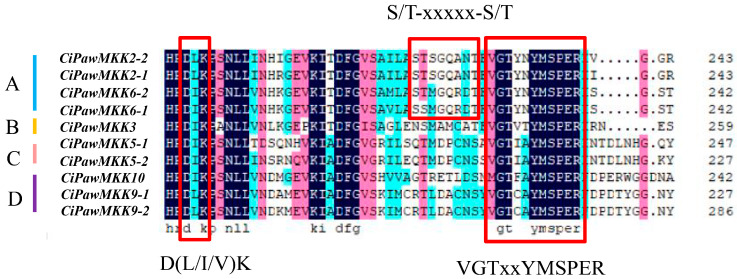
Multiple sequence alignment of MKK gene of pecan. The D(L/I/V)K and VGTxxYMSPER motifs specific to MKK genes and the S/T-xxxxx-S/T motifs contained in all group A genes are circled in red.

**Figure 3 ijms-23-15190-f003:**
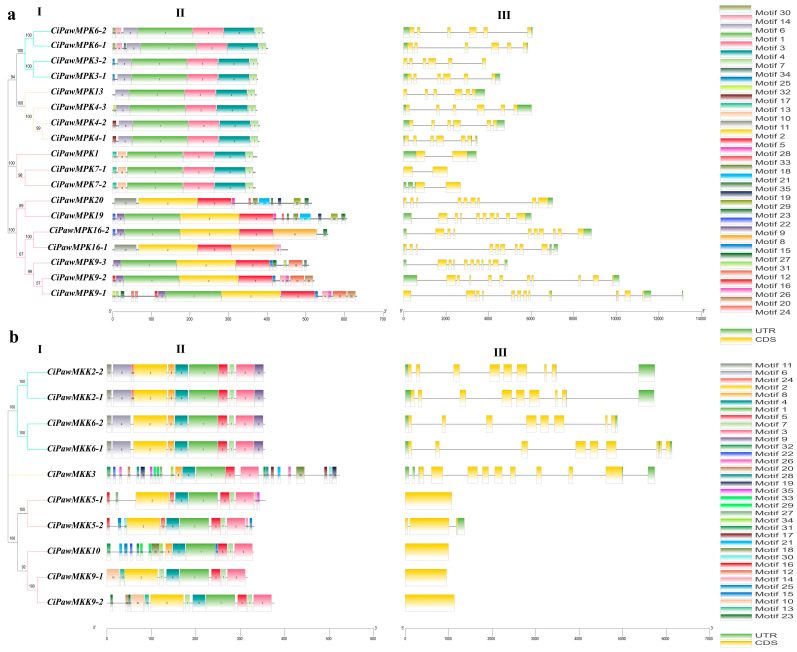
Evolutionary tree (**I**)**,** motifs (**II**)**,** and gene structure (**III**) analysis of MPK (**a**) and MKK (**b**) in pecan.

**Figure 4 ijms-23-15190-f004:**
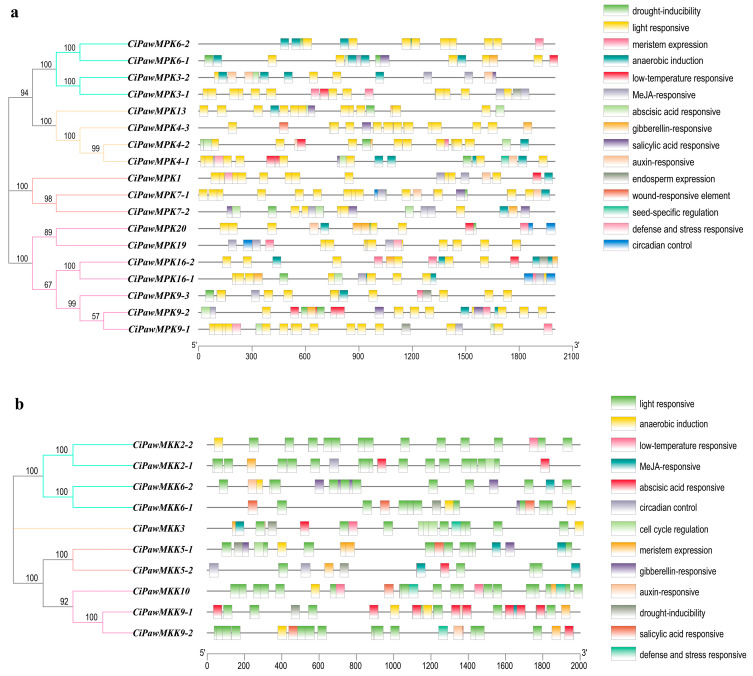
Cis-acting elements of MPK (**a**) and MKK (**b**) genes in pecan. Three types of cis-acting elements were found, including hormone-responsive elements, elements related to resistance to harsh environments and stress, and elements related to plant growth and development.

**Figure 5 ijms-23-15190-f005:**
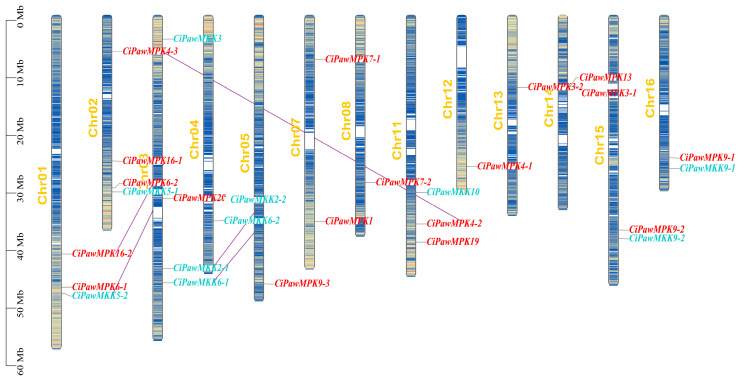
Distribution of MPK and MKK genes on pecan chromosomes. MPK is marked in red font, MKK genes are marked in light blue font, and pairs with interacting genes are connected by purple lines. The dense lines on each single chromosome show the distribution of all genes on this chromosome.

**Figure 6 ijms-23-15190-f006:**
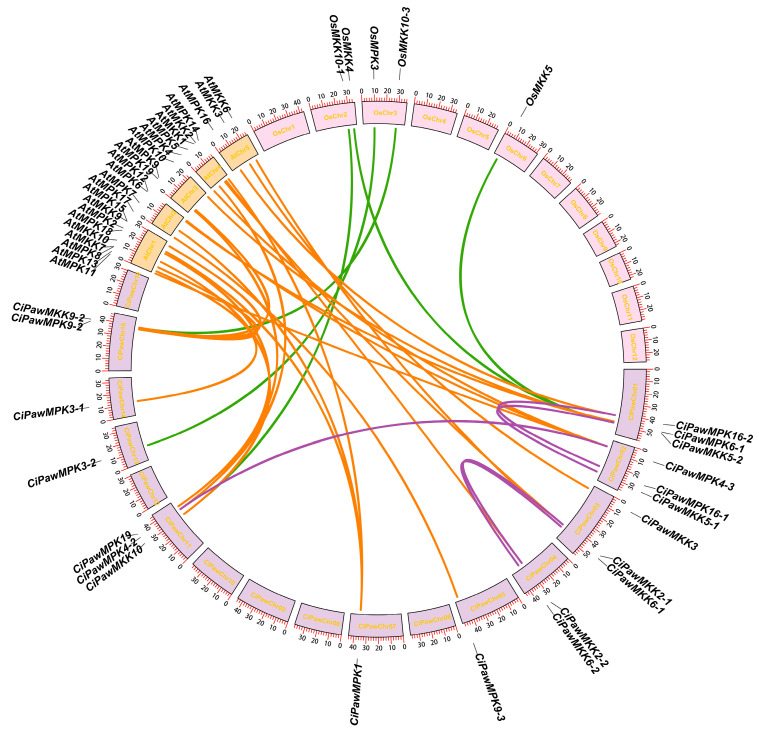
Analysis of duplication of MPK and MKK genes. Co-linear relationships between rice and pecan are shown as green arcs, between *Arabidopsis* and pecan as orange arcs, and between pecan itself as purple arcs.

**Figure 7 ijms-23-15190-f007:**
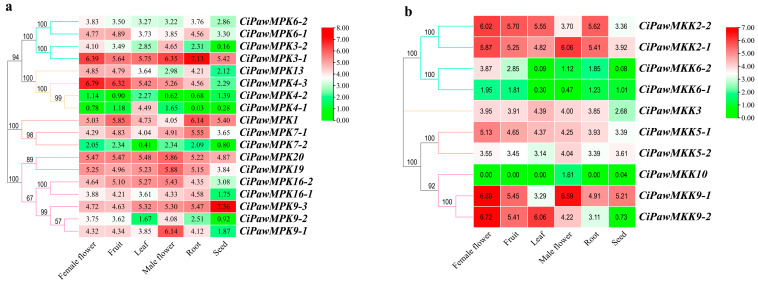
Expression of MPK (**a**) and MKK (**b**) genes in different organ samples of pecan.

**Figure 8 ijms-23-15190-f008:**
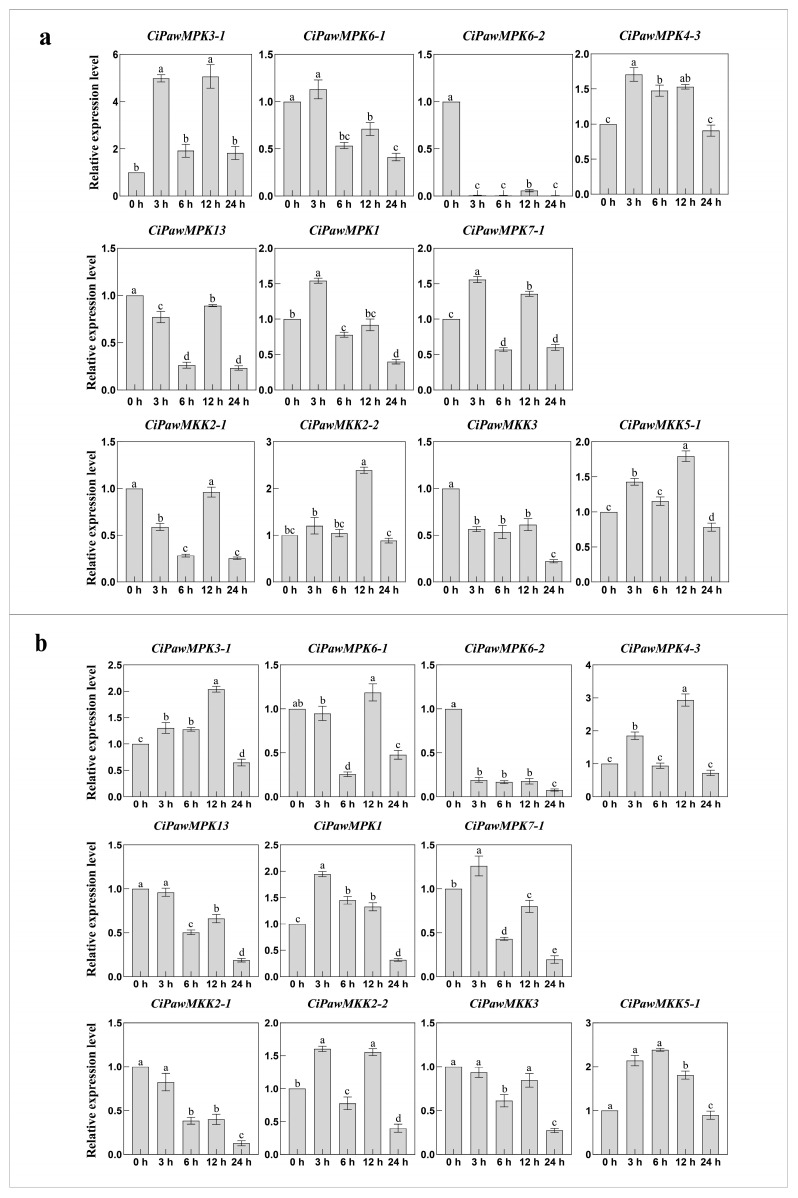
Gene expression patterns of MPK and MKK in pecan under SA (**a**) and ABA (**b**) treatments. The reference gene is the actin gene (*CiPaw.03G124400*). The horizontal coordinates indicate the different time points sampled and the vertical coordinates indicate the expression level of the genes. According to Duncan’s multiple range test, it was judged whether there was a significant difference in the expression levels at different time points (*p* < 0.05), and the difference results were denoted by lowercase letters. Error bars represent mean ± SE obtained from three biological replicates.

**Figure 9 ijms-23-15190-f009:**
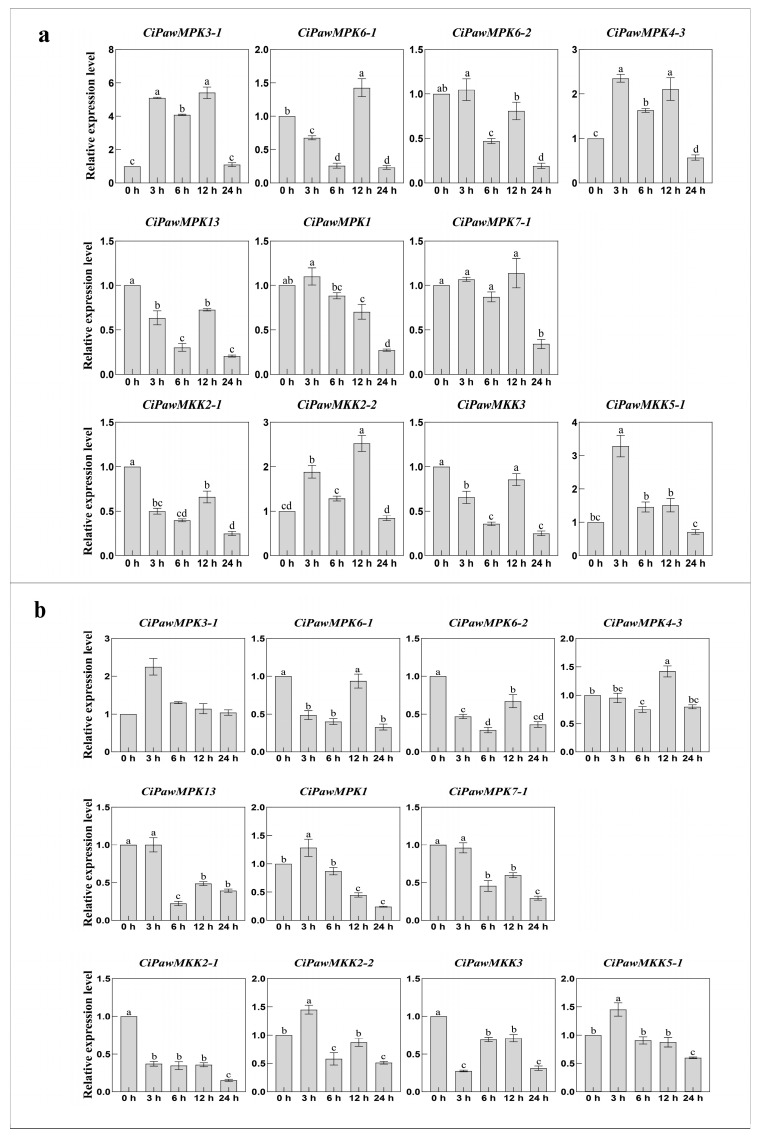
Gene expression patterns of MPK and MKK in pecan under MeJA (**a**) and 6-BA (**b**) treatments. The reference gene is the actin gene (*CiPaw.03G124400*). The horizontal coordinates indicate the different time points sampled and the vertical coordinates indicate the expression level of the genes. According to Duncan’s multiple range test, it was judged whether there was a significant difference in the expression levels at different time points (*p* < 0.05), and the difference results were denoted by lowercase letters. Error bars represent mean ± SE obtained from three biological replicates.

**Figure 10 ijms-23-15190-f010:**
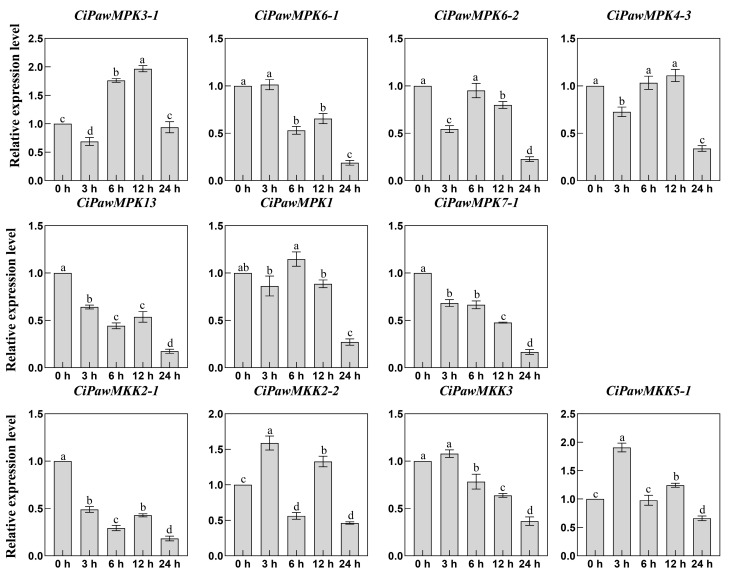
Gene expression patterns of MPK and MKK in pecan under H_2_O_2_ stress. The reference gene is the actin gene (*CiPaw.03G124400*). The horizontal coordinates indicate the different time points sampled and the vertical coordinates indicate the expression level of the genes. According to Duncan’s multiple range test, it was judged whether there was a significant difference in the expression levels at different time points (*p* < 0.05), and the difference results were denoted by lowercase letters. Error bars represent mean ± SE obtained from three biological replicates.

**Figure 11 ijms-23-15190-f011:**
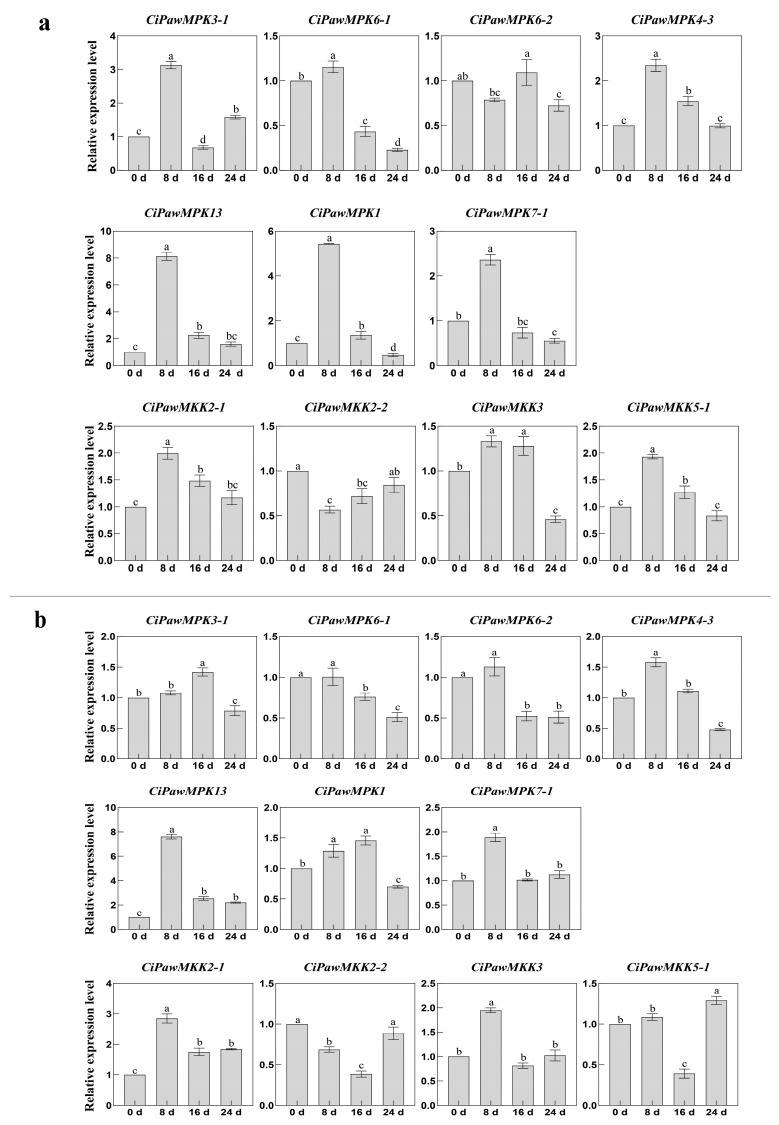
Gene expression patterns of MPK and MKK in pecan under NaCl (**a**) and 20% PEG (**b**) treatments. The reference gene is the actin gene (*CiPaw.03G124400*). The horizontal coordinates indicate the different time points sampled and the vertical coordinates indicate the expression level of the genes. According to Duncan’s multiple range test, it was judged whether there was a significant difference in the expression levels at different time points (*p* < 0.05), and the difference results were denoted by lowercase letters. Error bars represent mean ± SE obtained from three biological replicates.

**Table 1 ijms-23-15190-t001:** The *Ka*, *Ks* and *Ka/Ks* values of the collinear gene pairs of MPK and MKK in pecan, rice, and *Arabidopsis*.

Gene Pairs	*Ka*	*Ks*	*Ka/Ks*
*CiPawMPK4-2/CiPawMPK4-3*	0.0495	1.2872	0.0385
*CiPawMPK16-1/CiPawMPK16-2*	0.0175	0.2601	0.0672
*CiPaw.MKK2-1/CiPawMKK2-2*	0.0332	0.2276	0.1457
*CiPawMKK6-1/CiPawMKK6-2*	0.0225	0.2606	0.0865
*CiPawMKK5-1/CiPawMKK5-2*	0.0753	0.3572	0.2108
*CiPawMPK1/AtMPK2*	0.0414	2.9663	0.0139
*CiPawMPK4-3/AtMPK4*	0.0179	1.7541	0.0102
*CiPawMPK4-2/AtMPK5*	0.0386	1.4692	0.0262
*CiPawMPK6-1/AtMPK6*	0.0152	1.9237	0.0079
*CiPawMPK1/AtMPK7*	0.0681	—	—
*CiPawMPK9-2/AtMPK8*	0.0303	1.4757	0.0205
*CiPawMPK9-2/AtMPK9*	0.0251	1.8535	0.0135
*CiPawMPK6-1/AtMPK10*	0.0971	—	—
*CiPawMPK4-3/AtMPK11*	0.0362	2.6224	0.0138
*CiPawMPK4-3/AtMPK12*	0.0689	—	—
*CiPawMPK3-1/AtMPK13*	0.1766	—	—
*CiPawMPK1/AtMPK14*	0.0933	2.4142	0.0387
*CiPawMPK9-2/AtMPK15*	0.0458	1.0731	0.0427
*CiPawMPK16-2/AtMPK16*	0.0150	0.9426	0.0159
*CiPawMPK9-3/AtMPK17*	0.0808	—	—
*CiPawMPK19/AtMPK18*	0.0248	1.4943	0.0166
*CiPawMPK19/AtMPK19*	0.0298	1.6066	0.0186
*CiPawMKK2-1/AtMKK1*	0.2114	1.9369	0.1091
*CiPawMKK2-2/AtMKK2*	0.1628	1.6321	0.0997
*CiPawMKK3/AtMKK3*	0.0673	1.5936	0.0422
*CiPaw.MKK6-1/AtMKK6*	0.0304	1.1024	0.0275
*CiPawMKK9-2/AtMKK7*	0.2316	—	—
*CiPawMKK9-2/AtMKK9*	0.2139	—	—
*CiPawMKK10/AtMKK10*	0.2998	—	—
*CiPawMPK3-2/OsMPK3*	0.0909	—	—
*CiPawMKK5-2/OsMKK4*	0.1923	0.8339	0.2306
*CiPawMKK5-2/OsMKK5*	0.1852	0.9013	0.2055
*CiPawMKK10/OsMKK10-1*	0.3306	—	—
*CiPawMKK9-2/OsMKK10-3*	0.3284	1.0807	0.3038

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
