# Peer review of "Identification and Expression Analysis of MPK and MKK Gene Families in Pecan (Carya illinoinensis)"

_ijms, 2022, doi:10.3390/ijms232315190_

Round 1

Reviewer 1 Report

Reviewer’s comments

1.    In line 315, how redundant sequences were deleted from the raw data? Given the tool name and cite its reference.

2.    Give the taxonomical identification ID of pecan trees

3.    Mention the appropriate amount of collected tissue samples in line no 386

4.    Cite the reference article used for cDNA and RNA isolation 

5.    Give the primer sequence in Supplementary files

6.    Cite the reference article for “Detection of Enzyme Activity”

7.    Re-write the references according to the style of the Journal

8.    Explain the legend for Figure 8

9.    In line 68, what bioinformatics methods was used for the Identification, Nomenclature, and Protein Sequence Analysis of MPK and MKK Gene Families

10.Table 1and 2 can be added in the supplementary files.

11.Rephrase the figure 1 legend

12.Complete the kit name (â…¢ RT SuperMix ) and details in line no 390.

13. Mention the kit used for cDNA synthesis and its manufacturing details

14.Rephrase the sentence 393-395 

Reviewer 2 Report

The manuscript "Identification and Expression Analysis of MPK and MKK Gene Families in Pecan (Carya illinoinensis)" by Juan Zhao , Kaikai Zhu , Mengyun Chen , Wenjuan Ma , Junping Liu , Pengpeng Tan , Fangren Peng is a work to identify the expression of genes associated with important processes of resistance to stress. In its present form, the publication cannot be accepted for publication. It is necessary to eliminate the fundamental error of this material, affecting the entire manuscript.

The authors consider and describe their study as a study of tissue-specific expression. Meanwhile, not one part describes neither the methods for studying tissue-specific expression, nor the results of such studies. This misunderstanding is probably due to the fact that the authors do not take into account that they studied not tissue-specific, but organ-specific expression. In each organ, as the authors probably know, there are many different tissues, while they also have different ploidy, which also affects the conclusions that could be drawn from this. But we can say for sure that any organ has an epidermis (which contains both stomatal cells, epidermal cells and cells of various hairs), parenchyma is observed between the words of the epidermis, there are also cells of mechanical and vascular tissues (xylem, phloem), also specialized cells, for example, ideoblasts . These are tissues, they are in different organs. There are also various meritematic tissues and, for example, tissues of the root cap, which are associated with gravitropism.

It is likely that the paper should be rewritten to describe the work as an organ-specific expression analysis.

On the other hand, the design of the manuscript does not meet the requirements of the journal (not only the design and font).

So there is no conclusion. Also, at the end of the introduction, it is always necessary to clearly and unambiguously describe the goals and objectives of the study, and not describe what has been done.

The manuscript contains important information, but it is impossible to read it. Even when the drawings are enlarged, it is almost impossible to make out the inscriptions.

I think that if this work is carefully reworked, it can be re-examined and it can come out.

Round 2

Reviewer 2 Report

Manuscript "Identification and Expression Analysis of MPK and MKK Gene Families in Pecan (Carya illinoinensis)" by

Juan Zhao, Kaikai Zhu, Mengyun Chen, Wenjuan Ma, Junping Liu, Pengpeng Tan, Fangren Peng has been significantly improved, major and minor changes have been made. Erroneous terms have been corrected. Conclusion added. The article is properly formatted and can be published.